# Development and validation protocol for an instrument to measure household water insecurity across cultures and ecologies: the Household Water InSecurity Experiences (HWISE) Scale

Sera L Young,[1] Shalean M Collins,[2] Godfred O Boateng,[2] Torsten B Neilands,[3] Zeina Jamaluddine,[4] Joshua D Miller,[2] Alexandra A Brewis,[5] Edward A Frongillo,[6] Wendy E Jepson,[7] Hugo Melgar-Quiñonez,[8] Roseanne C Schuster,[5] Justin B Stoler,[9] Amber Wutich,[5] on behalf of the HWISE Research Coordination Network

For numbered affiliations see end of article.

**Correspondence to**
Dr Sera L Young;
sera.young@northwestern.edu

## ABSTRACT

**Introduction** A wide range of water-related problems contribute to the global burden of disease. Despite the many plausible consequences for health and well-being, there is no validated tool to measure individual- or household-level water insecurity equivalently across varying cultural and ecological settings. Accordingly, we are developing the Household Water Insecurity Experiences (HWISE) Scale to measure household-level water insecurity in multiple contexts.

**Methods and analysis** After domain specification and item development, items were assessed for both content and face validity. Retained items are being asked in surveys in 28 sites globally in which water-related problems have been reported (eg, shortages, excess water and issues with quality), with a target of at least 250 participants from each site. Scale development will draw on analytic methods from both classical test and item response theories and include item reduction and factor structure identification. Scale evaluation will entail assessments of reliability, and predictive, convergent, and discriminant validity, as well as the assessment of differentiation between known groups.

**Ethics and dissemination** Study activities received necessary ethical approvals from institutional review bodies relevant to each site. We anticipate that the final HWISE Scale will be completed by late 2018 and made available through open-access publication. Associated findings will be disseminated to public health professionals, scientists, practitioners and policymakers through peer-reviewed journals, scientific presentations and meetings with various stakeholders. Measures to quantify household food insecurity have transformed policy, research and humanitarian aid efforts globally, and we expect that an analogous measure for household water insecurity will be similarly impactful.

## Strengths and limitations of this study

► This study is based on rigorous, multidisciplinary formative research on water insecurity by anthropologists, geographers, nutritionists, statisticians and epidemiologists, among others.
► Data on household water insecurity experiences are being collected in 28 sites across four continents by local partners in widely varying ecological and cultural settings.
► Analytic methods from both classical test and item response theories will be used to develop and evaluate the eventual scale.
► The Household Water Insecurity Experiences Scale will be validated for assessing water insecurity in low-income and middle-income countries. Additional scale assessments necessary for validation in high-income countries are planned.

## INTRODUCTION

Water insecurity, the inability to 'access and benefit from affordable, adequate, reliable and safe water for well being and a healthy life',[1] has manifold adverse effects on physical[2 3] and psychosocial health[4 5]; undermines productivity[6]; triggers and perpetuates domestic, social and political tensions and conflicts[7 8]; and reinforces environmental and social inequities.[9] Water insecurity has been shown to co-occur with food insecurity, malnutrition and communicable diseases and to produce syndemics, or systemically exacerbating epidemics,[10 11] much like food insecurity and HIV.[12] Furthermore, water insecurity is projected to worsen in many regions due to climate change and increased inequalities in resource distribution.[9]

However, we do not know how many households or individuals globally are affected by water insecurity. Estimates of available surface water derived from satellite imagery suggest that 4 billion people worldwide experience severe water scarcity for at least 1 month of every year,[13] and this is likely an underestimation given issues with infrastructure and access. Additionally, chronic flooding[14] and poor water quality[15] mean that many more individuals are experiencing water insecurity. Currently, measures of water at the national, regional and community levels are used and are referred to as indicators of water scarcity, water poverty or water security.[16–19] These measures do not capture the range and granularity of how households experience water insecurity, including factors such as perceptions of quality,[20] instances of water excess[21] or perceived consequences for psychosocial[4 5] and physical health and well-being.[22] Furthermore, while household-level scales to measure water insecurity have been developed for several sites, for example, in the USA,[23] Bolivia,[4] Uganda,[24] Ethiopia[5] and Kenya,[10] their comparability, comprehensiveness and applicability to other sites have not been systematically investigated or validated.

As such, a comprehensive, validated scale to measure the experiences of household or individual water insecurity would enable researchers, practitioners and policymakers to: improve estimates of water insecurity prevalence, identify factors that shape this phenomenon, recognise direct consequences of water insecurity, understand how to more effectively distribute resources, evaluate the impacts and cost-effectiveness of interventions and monitor progress towards the Sustainable Development Goals.[25] Indeed, in March 2018, the UN's High-Level Panel on Water cited lack of data on water in many parts of the world as a major challenge, and the need for better data on water as one of nine priority actions.[26] Given that measures of household food insecurity (eg, Latin American and Caribbean Food Security Scale,[27] Household Food Insecurity Access Scale,[28] Food Insecurity Experience Scale[29]) have proven vital to implementation and evaluation of policy and programmes,[30–32] development of an analogous household water insecurity scale is overdue and urgently needed, particularly for assessing water insecurity in low-income and middle-income settings where household water problems tend to be most pronounced and frequent.

Therefore, our objective is to develop and validate the first household water insecurity scale with broad applicability across low- and middle-income settings. The Household Water Insecurity Experiences (HWISE, pronounced *H-wise*) Research Coordination Network (RCN) was formed to facilitate the multicountry, collaborative data collection process required to validate the planned tool ('the HWISE Scale'). The HWISE RCN brings together a large team of anthropologists, geographers, public health practitioners, physicians, epidemiologists, statisticians, sociologists, nutritionists, inter alia, all of whom have experience with water insecurity, food insecurity and/or

scale development across a wide array of settings (http://www.hwise.org).

## METHODS AND ANALYSIS
### Phase 1: item development
#### Domain specification
Specifying the domains for a scale is the first step in item development (table 1, 1.1).[33 34] The boundary of the domain of water insecurity, that is, the underlying construct that the scale will attempt to measure, was based on extensive literature review[1] and draws on the team's expertise in water insecurity, for example.[4 8 23 35] We define water insecurity as the condition where 'affordability, reliability, adequacy, and safety [of water] is significantly reduced or unattainable so as to threaten or jeopardize well-being'.[1]

A best practice is to clearly articulate subdomains of the eventual scale, if they are known.[34 36] Although some subdomains of water insecurity have been proposed,[1 5 11 37] there is currently no consensus in the literature. Therefore, we will assess subdomains during the analytic phase.

#### Item generation
Candidate scale items were identified deductively, based on an extensive literature review of items used in prior site-specific household water insecurity scales[1] (table 1, 1.2). This includes team members' prior work in *colonias* in the USA-Mexico boderlands[23]; a squatter settlement in Cochabamba, Bolivia[4]; in rural, periurban and urban households in Kenya[10]; and elsewhere, including rural areas in Ethiopia[5] and Uganda.[24] Initial items include experiences of water insecurity that have consequences for psychosocial and physical health, nutrition, impacts on livelihoods and household economy, and agriculture (online supplementary file 1).

Each question is phrased to elicit experiences within the prior 4 weeks or month (ie, 'In the last four weeks, how frequently have…'). This recall period was systematically determined using the Delphi method of consensus building with international and local experts in water insecurity, food insecurity and scale development and by comparing the responses in this recall period to a prospective daily recall of water insecurity experiences.[10] Items were ordered by what we expected to be increasing severity of water insecurity across access, reliability, adequacy and safety. Response options are 'never' (0 times), 'rarely' (1–2 times), 'sometimes' (3–10 times), 'often' (11–20 times), 'always' (more than 20 times), 'not applicable', 'don't know' or refused. Response intervals were also determined using the Delphi method.[10]

The initial set of 32 items is referred to as 'Module Version 1'. This set of items was modified slightly in August 2017 (see 'Mid-study Evaluations' under Phase 3) based on feedback received from consortium members, survey implementers and other water security experts during a 3-day conference at Northwestern University. Modifications included slight rephrasing of 18 items to improve

**Table 1** Overview of planned methods and analyses for the development of the HWISE Scale*

| Scale development activity | Procedures |
| --- | --- |
| **Phase 1: item development** | |
| 1.1 Domain specification | Literature review. |
| 1.2 Item generation | Literature review and Delphi methodology. |
| 1.3 Content validity | By target population: two styles of cognitive interviews were used in the first eight sites, building on Delphi methodology. |
| 1.4 Face validity | Pretesting and debriefing with enumerators at each site. |
| **Phase 2: scale development** | |
| 2.1 Data collection | Enumerator training and survey implementation. |
| 2.2 Item reduction | We will drop items with cumulative missing cases >30% (ie, 'don't know', 'non-applicable' or true missing responses) in any one site. |
| | We will assess the performance of each item's variation with other items in the scale using a correlation matrix; items with very low (<0.30) interitem correlation coefficients and very low (<0.30) item-total correlation coefficients across multiple sites will be considered for deletion, as will items that misfit the model, that is, with residual correlations >0.20. |
| | Item reduction in Rasch paradigm: item severity and item discrimination test. |
| 2.3 Identify factor structure | We will use factor analysis across multiple sites to test for factor structure; items with very low factor loadings (<0.30), split factor loadings (high factor loadings (>0.50) in two domains) and high residual variances will be considered for deletion. |
| 2.4 Assess measurement equivalence | We will use multigroup confirmatory analysis (a form of measurement invariance) on data from multiple sites to test for exact invariance in the hypothetical scale; invariance will be assessed in terms of factor structure (configural model), factor loadings (matric model), mean intercepts (scalar model) and factor means (strict model). |
| | We will use confirmatory factor analysis alignment optimisation to estimate the group-specific factor means and variances of scale items across all sites; it assesses approximate invariance of scale items across multiple sites. |
| **Phase 3: scale evaluation** | |
| 3.1 Score scale items | Finalised scale items will be used in their unweighted form as sum scores or in weighted form as factor scores. |
| 3.2 Assess reliability (internal consistency) of scale items | We will use Cronbach's alpha and the Rasch reliability statistic to test the internal consistency of the scale items within each site and aggregated across sites. |
| 3.3 Assess scale validity | We will measure predictive, convergent and discriminant validity of the final scale items using criteria that were selected based on their strong theoretical relevance in the water insecurity literature; tests of water insecurity differences between 'known groups' will also be performed. |

*Adapted from ref [34].

comprehension by participants and to elicit experiences related to water overabundance, two questions were added in an effort to capture cultural components of water and six items were eliminated for being too rare or idiosyncratic. The resultant set of 28 items is referred to as 'Module Version 2' (online supplementary file 1).

### Content validity
Content validity (ie, if items adequately measure the domain of interest; table 1, 1.3) was assessed in the first eight sites through cognitive interviews with 12 purposively selected individuals. Participants were asked to 'think aloud' or 'tell [the enumerator] about' their understanding of each of the water insecurity items as they completed the pilot survey. Interviewers recorded

any issues and probed in detail on each as participants responded to the items.[38] This process built on the Delphi methodology used to develop the Kenya-specific scale.[10]

### Face validity
Face validity, also part of item development, is assessed at each site (table 1, 1.4). First, the survey is translated from English into the language(s) of implementation and then back-translated. Then, enumerators, the predominance of whom are recruited from the target population, pretest surveys with one another to ensure that questions are appropriate to the setting, that the concept of water insecurity is understood and that translations are consistent with local dialects, that is, that they are linguistically and culturally appropriate translations.[29]

Site leads debrief enumerators after each day of data collection and record all the details as project field notes to further ensure face validity. Debriefs are centred on experiences in the community, survey questions that are difficult to administer and any other problems encountered. At the end of data collection for the site, enumerators engage in a final debrief, and in some cases, use a semistructured survey that pulls the same information from across the entire site. Site leads are also interviewed at the end of study activities by members of the HWISE RCN regarding their experiences with project implementation, perceptions of questions by enumerators and participants and any additional topics that should be included or excluded in the final survey. These debriefing interviews with site leads will provide additional feedback to iteratively improve training and item refinement.

## PHASE 2: SCALE DEVELOPMENT
### Data collection
#### Sites
Cross-sectional surveys were initially planned for six sites that would leverage investigators' active research: Bangladesh, Brazil, Guatemala, Kenya, Nepal and Tajikistan; that is, they were selected out of convenience. Subsequently, 22 more were added because additional sites would allow us to test the instrument across more heterogeneous cultural and geographic settings (figure 1), permit an iterative analysis of the instrument (compared with 'Mid-study evaluation') and make a number of statistical analyses possible (table 2).[39] These additional 22 sites were added by soliciting collaborators from professional networks across academic institutions as well as non-governmental and governmental organisations using convenience sampling. In selecting sites, we sought maximal heterogeneity in region of the world, infrastructure (eg, urban and rural, formal and informal settlements) and problems with water (eg, flooding, drought, chronic scarcity and intermittent supplies). We also considered cost and feasibility of timely implementation.

#### Participant selection
To participate, individuals must be 16 or 18 years of age or older (depending on age of consent at each site), identify themselves to the interviewer as being knowledgeable about water acquisition and use within their households and consent to participate. Participants are not remunerated for participation in the survey.

The target sample size at each site is 250 individuals. We consider this sample size as the minimum needed for assessing the magnitude of correlation between the observed variables and associated factor(s) and obtaining a sample pattern that is stable and approximates the population pattern.[40] If sites cannot achieve the target sample size, variation of estimated statistics will be reviewed to determine if the data can be included in the final validation of the scale.

The preferred sampling strategy for the study is random sampling of mutually exclusive and exhaustive categories of participants in areas of known high, moderate and low

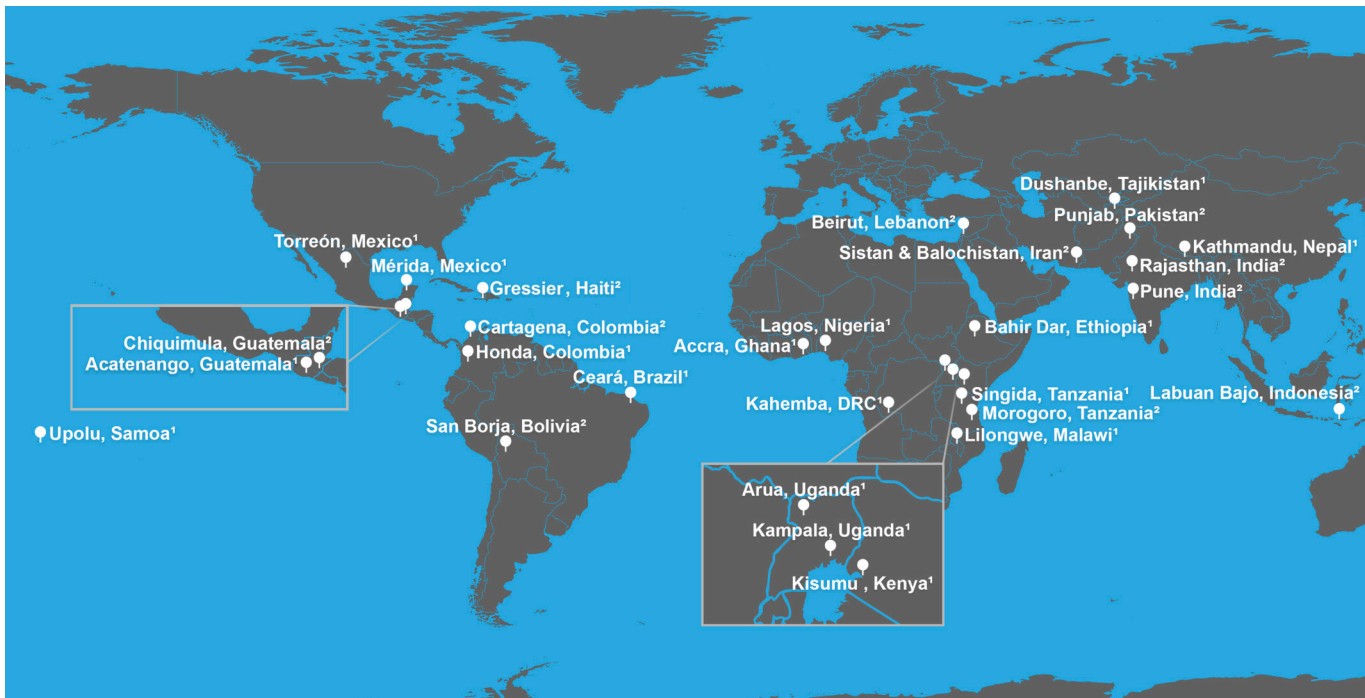

**Figure 1** Map of HWISE study sites. [1] Sites using Module Version 1; [2] Sites using Module Version 2. Image credit: Frank Elavsky, Northwestern University Information Technology, Research Computing Services. HWISE, Household Water Insecurity Experiences.

**Table 2** Characteristics of HWISE sites for scale development by region

| World Bank region | Site (Module Version) | Primary sources of drinking water, % | Köppen climate classification at site (Köppen code)* | GNI per capita (USD)† | National income classification‡ | Urbanicity of site |
|---|---|---|---|---|---|---|
| Africa | Accra, Ghana (1) | Bagged/sachet water, 86.0. Borehole/tubewell, 5.7. Other, 8.3. | Equatorial, winter dry (Aw). | 1380 | Lower middle income. | Urban |
| | Lagos, Nigeria (1) | Bagged/sachet water, 48.9. Borehole/tubewell, 34.7. Other, 16.4. | Equatorial, winter dry (Aw). | 2450 | Lower middle income. | Urban |
| | Kahemba, DRC (1) | Surface water, 99.7. Other, 0.3. | Equatorial, winter dry (Aw). | 420 | Low income. | Rural |
| | Bahir Dar, Ethiopia (1) | Unprotected dug well, 25.1. Rainwater collection, 20.9. Standpipe, 13.5. Surface water, 13.5. Protected dug well, 12.4. Unprotected spring, 10.0. Other, 4.6. | Warm temperature, winter dry, warm summer (Cwb). | 660 | Low income. | Rural |
| | Singida, Tanzania (1) | Standpipe, 48.6. Unprotected dug well, 17.4. Borehole/tubewell, 12.9. Other, 12.8. Unprotected spring, 8.3. | Equatorial, winter dry (Aw). | 900 | Low income. | Rural |
| | Lilongwe, Malawi (1) | Standpipe, 45.4. Piped water, 42.1. Other, 12.5. | Warm temperature, winter dry, hot summer (Cwa). | 320 | Low income. | Periurban |
| | Arua, Uganda (1) | Protected dug well, 64.8. Unprotected spring, 19.6. Other, 15.6. | Equatorial, winter dry (Aw). | 660 | Low income. | Rural |
| | Kisumu, Kenya (1) | Surface water, 17.4. Borehole/tubewell, 16.2. Rainwater, 13.8. Piped water, 11.3. Standpipe, 10.9. Protected dug well, 10.1. Unprotected dug well, 7.7. Unprotected spring, 6.1. Other, 6.5. | Equatorial, fully humid (Af). | 1380 | Lower middle income. | Rural |
| | Kampala, Uganda (1) | Standpipe, 68.3. Other, 21.1. Unprotected dug well, 10.6. | Equatorial, winter dry (Aw). | 660 | Low income. | Urban |
| | Morogoro, Tanzania (2) | Standpipe, 70.7. Other, 29.3. | Equatorial, winter dry (Aw). | 900 | Low income. | Urban and periurban |
| East Asia and Pacific | Upolu, Samoa (1) | Piped water, 81.4. Standpipe, 12.5. Other, 6.1. | n.d. | 4100 | Upper middle income. | Urban and periurban |
| | Labuan Bajo, Indonesia (2) | Bagged/sachet water, 36.9 .Protected spring, 12.9. Piped water, 10.0. Tanker truck, 9.7. Standpipe, 9.3. Protected dug well, 6.5. Borehole/tubewell, 5.7. Other, 9.0. | Equatorial, fully humid (Af). | 3400 | Lower middle income. | Urban |
| Europe and Central Asia | Dushanbe, Tajikistan (1) | Piped water, 58.2. Standpipe, 24.0. Tanker truck, 9.3. Other, 8.5. | Warm temperature, summer dry, warm summer (Csb). | 1110 | Lower middle income. | Urban |

Continued

**Table 2** Continued

| World Bank region | Site (Module Version) | Primary sources of drinking water, % | Köppen climate classification at site (Köppen code)* | GNI per capita (USD)† | National income classification‡ | Urbanicity of site |
|---|---|---|---|---|---|---|
| Latin America and the Caribbean | Ceará, Brazil (1) | Piped water, 59.5. Protected dug well, 33.9. Bottled water, 5.5. Other, 1.1. | Equatorial, summer dry (As). | 8840 | Upper middle income. | Urban |
| | Mérida, Mexico (1) | Bagged/sachet water, 50.0. Other, 33.6. Piped water, 14.4. Other, 2.0. | Arid steppe, hot arid (BSh). | 9040 | Upper middle income. | Urban |
| | Acatenango, Guatemala (1) | Piped water, 38.4. Standpipe, 31.3. Tanker truck, 16.2. Other, 14.1. | Warm temperature, winter dry, warm summer (Cwb). | 3790 | Lower middle income. | Periurban |
| | Honda, Colombia (1) | Piped water, 74.5. Standpipe, 20.4. Other, 5.1. | Equatorial, fully humid (Af). | 6320 | Upper middle income. | Periurban |
| | Torreón, Mexico (2) | Bottled water, 70.2. Piped water, 27.0. Other, 2.8. | Arid steppe, hot arid (BSh). | 9040 | Upper middle income. | Urban |
| | San Borja, Bolivia (2) | Standpipe, 41.6. Tanker truck, 19.3. Other, 10.1. Borehole/tubewell, 8.0. Piped water, 7.6. Rainwater collection, 6.7. Bottled water, 6.7. | Equatorial, monsoonal (Am). | 3070 | Lower middle income. | Rural |
| | Chiquimula, Guatemala (2) | Piped water, 65.0. Unprotected spring, 15.3. Standpipe, 12.7. Other, 7.0. | Equatorial, monsoonal (Am). | 3790 | Lower middle income. | Rural |
| | Gressier, Haiti (2) | Standpipe, 26.8. Small water vendor, 14.1. Bagged/sachet water, 13.1. Other, 10.9. Bottled water, 10.7. Borehole/tubewell, 9.3. Protected dug well, 7.9. Tanker truck, 7.2. | Equatorial, winter dry (Aw). | 780 | Low income. | Periurban |
| | Cartagena, Colombia (2) | Piped water, 46.2. Standpipe, 34.6. Other, 12.4. Small water vendor, 6.8. | Equatorial, winter dry (Aw). | 6320 | Upper middle income. | Urban |
| Middle East and North Africa | Beirut, Lebanon (2) | Small water vendor, 54.5. Bottled water, 39.7. Other, 5.8. | Warm temperature, summer dry, hot summer (Csa). | 7680 | Upper middle income. | Urban |
| | Sistan and Balochistan, Iran (2) | Small water vendor, 48.0. Other, 30.1. Piped water, 21.9. | Arid, desert, hot arid (BWh) | 5470 | Upper middle income. | Urban, periurban and rural |

Continued

**Table 2** Continued

| World Bank region | Site (Module Version) | Primary sources of drinking water, % | Köppen climate classification at site (Köppen code)* | GNI per capita (USD)† | National income classification‡ | Urbanicity of site |
|---|---|---|---|---|---|---|
| South Asia | Kathmandu, Nepal (1) | Bottled water, 49.8. Piped water, 31.2. Tanker truck, 10.7. Other, 8.3. | Warm temperature, winter dry, hot summer (Cwa). | 730 | Low income. | Urban |
| | Pune, India (1) | Piped water, 89.4. Other, 10.6. | Equatorial, winter dry (Aw). | 1680 | Lower middle income. | Urban |
| | Punjab, Pakistan (2) | Standpipe, 26.6. Borehole/tubewell, 23.2. Piped water, 15.9. Rainwater collection, 14.2. Small water vendor, 10.3. | Arid, desert, hot arid (BWh). | 1510 | Lower middle income. | Rural and periurban |
| | Rajasthan, India (2) | Tanker truck, 55.2. Borehole/tubewell, 26.2. Other, 13.4. Piped water, 5.2. | Arid steppe, hot arid (BSh). | 1680 | Lower middle income. | Urban |

*Köppen climate classification predicted using Scenario A1F1 for 2001–2025, projected to 31 December 2020 used for reference point (ESRI, ArcGIS).
†Gross National Income in USD from World Bank classification, data from 2017.
‡Income Classification from World Bank, data from 2017.
DRC, Democratic Republic of the Congo; n.d., no data.

water insecurity. In standalone HWISE sites, participant selection follows a simple randomised or cluster-randomised sampling strategy (table 3). In several sites, however, the HWISE survey is administered as part of a larger ongoing project with a predetermined survey design (eg, in Singida, Tanzania: NCT02761876; Kahemba, Democratic Republic of Congo: NCT03157336), such that simple random sampling is not possible.

Sites with simple randomised sampling employ a random-walk sampling method.[41] With the simple randomised sampling strategy, a random number generator (eg, dice or random number generating application) with set parameters (ie, less than 20, less than 30 and so on) determines which households to survey (and, if needed, the direction of the random walk). Surveys are administered to each household corresponding to the random number, such that a random draw of the number 3 indicates that every third household should be sampled. For sites using a cluster-randomised sampling strategy, the region is first mapped using a grid or satellite imagery (eg, Google Maps) to identify population density based on the number of habitable structures. Clusters are selected from this grid, and households within clusters are randomly sampled in proportion to structure or population density using a random number generator, similar to the simple randomised sampling strategy. Cluster randomisation is preferred, but simple random sampling is used when cluster data are not available, typically in sparsely populated settings.

### Participant involvement

Although formative work drew on ethnographic research that included participant involvement and the idea to develop this scale came from experiences with participants in Kenya,[1] no participants were involved in developing the actual protocol. Participant involvement (eg, cognitive interviewing; table 1, 1.3) began with refinement of survey items once the initial list was created. Participants were not involved in developing plans for the design or implementation of the study, and participants will not be involved in the interpretation of results or write-up of the manuscript. Although identifiable data were not collected in most sites, we plan to summarise our findings in site-specific summary reports that site investigators will disseminate to communities in which the data were collected. The final scale and other findings will be made available via open-access publication and be publicised through public relations and media outreach at our respective institutions.

### Training

An HWISE training manual was developed to provide guidance on implementation.[42] This manual outlines preferred sampling strategy, minimum sample size, instructions for collecting data and choosing unique participant identification numbers and detailed information explaining the rationale for each water insecurity

**Table 3** Overview of data collection activities at each HWISE study site

| World Bank region | Site | Module Version | Implementing partners | Month(s) and year of data collection | Sample size | Female respondents, % | Season of data collection | Language(s) of data collection | Sampling strategy | Data collection method (software) | Details of larger study; supplementary data collected | IRB of record | Cognitive interviewing (Y/N) |
|---|---|---|---|---|---|---|---|---|---|---|---|---|---|
| Africa | Accra, Ghana | 1 | University of Miami, Delaware State University, Ghana Water Company and Northwestern University. | June 2017 | 229 | 78.2 | Rainy season. | English. | Simple random. | Tablet (ODK). | Standalone. | University of Miami; Delaware University, reliant on Northwestern University and Ghana Water Company. | Yes |
| | Lagos, Nigeria | 1 | College of Medicine at the University of Lagos and Northwestern University. | June–August 2017 | 239 | 73.5 | Rainy season. | English, Yoruba and Pidgin. | Multistage random. | Paper, entered into database (Enketo). | Standalone; adolescent menstrual hygiene. | Northwestern University & University of Lagos | Yes |
| | Kahemba, DRC | 1 | Oregon Health Sciences University, Michigan State University and Institut National de Recherche Biomedicale. | June–September 2017 | 392 | 65.6 | Dry season. | Kikongo and Lingala. | Cluster randomised control trial. | Paper/tablet hybrid (ODK). | NCT03157336: toxicodietary and genetic determinants of susceptibility to neurodegeneration. | Oregon Health Sciences University & Ministry of Health, DRC | No |
| | Bahir Dar, Ethiopia | 1 | Oregon State University, Emory University and Emory Ethiopia. | July–August 2017 | 259 | 100 | Rainy season. | Amharic. | Stratified random. | Tablet (KoboToolbox). | NCT03075436: the impact of enhanced demand-side sanitation and hygiene promotion on sustained behaviour change and health in Ethiopia. | Amhara Regional Health Bureau, Emory University; Oregon State University, reliant on Northwestern University | No |
| | Singida, Tanzania | 1 | Cornell University and Northwestern University. | July–August 2017 | 1006* | 56.7 | Dry season. | Swahili. | Purposive, community led. | Tablet (ODK). | NCT02761876: Singida nutrition and agroecology project. | Cornell University | No |
| | Lilongwe, Malawi | 1 | Georgia State University. | July 2017 | 302 | 86.8 | Neither rainy nor dry season. | Chichewa and English. | Cluster random. | Tablet (ODK). | Standalone. | Georgia State University | No |
| | Kisumu, Kenya | 1 | Pamoja Community Based Organisation and Northwestern University. | July 2017 | 247 | 81.3 | Neither rainy nor dry season. | Luo, Swahili and English. | Simple random. | Tablet (ODK). | Standalone; moringa. | Northwestern University and African Medical Research Foundation. | Yes |
| | Arua, Uganda | 1 | Michigan State University. | August–September 2017 | 250 | 85.6 | Rainy season. | Lugbara and English. | Cluster random. | Paper, entered into database (Enketo). | Standalone. | Michigan State University, reliant on Northwestern University. | No |
| | Kampala, Uganda | 1 | T-Group Kampala, University of Amsterdam and Makerere University. | August 2017 | 246 | 69.1 | Dry season. | Luganda and English. | Purposive. | Paper, entered into database (Enketo). | Standalone. | Northwestern University and T-Group Kampala. | No |
| | Morogoro, Tanzania | 2 | Workman Consulting and Northwestern University. | March–May 2018 | 300 | 78.3 | Rainy season. | Swahili. | Cluster random. | Paper, entered into database (Enketo). | Standalone; water for sanitation and hygiene. | Northwestern University and Sokoine University of Agriculture. | No |
| East Asia and Pacific | Upolu, Samoa | 1 | Yale University. | April 2018–present | 176† | 63.4† | Across multiple seasons. | Samoan. | Purposive. | Tablet (REDCap). | NIH R01HL093093: integrated cellular, mouse and human research on a novel missense variant influencing adiposity in Samoans. | Yale University. | No |
| | Labuan Bajo, Indonesia | 2 | University of the West of England. | May 2018 | 279 | 44.8 | Dry season. | Indonesian. | Cluster random. | Tablet (ODK). | Standalone. | Exempt. | No |
| Europe and Central Asia | Dushanbe, Tajikistan | 1 | Arizona State University and M-Vector. | July–August 2017 | 225 | 73.3 | Dry season. | Tajik and Russian. | Cluster random. | Tablet (CSPro). | Global Ethnohydrology Study—ASU. | Arizona State University. | Yes |

Continued

**Table 3** Continued

| World Bank region | Site | Module Version | Implementing partners | Month(s) and year of data collection | Sample size | Female respondents, % | Season of data collection | Language(s) of data collection | Sampling strategy | Data collection method (software) | Details of larger study; supplementary data collected | IRB of record | Cognitive interviewing (Y/N) |
|---|---|---|---|---|---|---|---|---|---|---|---|---|---|
| Latin America and the Caribbean | Ceará, Brazil | 1 | Texas A&M University. | March 2017–February 2018 | 254 | 70.2 | Neither rainy nor dry season. | Portuguese. | Cluster random. | Paper, entered into database (Enketo). | NSF1560962: urban water provisioning systems and household water security. | Texas A&M University. | No |
| | Mérida, Mexico | 1 | Michigan State University. | July–August 2017 | 250 | 63.4 | Dry season. | Spanish. | Cluster random. | Paper, entered into database (Enketo). | Standalone. | Michigan State University and Northwestern University. | No |
| | Honda, Colombia | 1 | Pontificia Universidad Javeriana and Northwestern University. | August 2017 | 196 | 63.6 | Rainy season. | Spanish. | Cluster random. | Tablet (ODK). | Standalone. | Northwestern University and Pontificia Universidad Javeriana. | No |
| | Acatenango, Guatemala | 1 | Arizona State University. | September–October 2017 | 101 | 93.0 | Dry season. | Spanish. | Cluster random. | Paper, entered into database (Excel). | Global Ethnohydrology Study—ASU. | Arizona State University. | No |
| | San Borja, Bolivia | 2 | Pennsylvania State University. | November–December 2017 | 247 | 58.6 | Dry season. | Spanish. | Simple random. | Paper, entered into database (Excel). | Standalone. | Northwestern University. | No |
| | Chiquimula, Guatemala | 2 | McGill University and Action Against Hunger-Guatemala. | January–February 2018 | 314 | 86.6 | Middle/end of dry season. | Spanish. | Systematic random. | Tablet (ODK). | Standalone. | Action Against Hunger-Guatemala and Northwestern University. | No |
| | Gressier, Haiti | 2 | University of Florida. | February–March 2018 | 292 | 98.6 | Dry season. | Creole. | Stratified random. | Tablet (REDCap). | Standalone; perceived water quality and disease. | University of Florida. | No |
| | Torreón, Mexico | 2 | Texas A&M University. | April 2018 | 249 | 73.1 | Middle/end of dry season. | Spanish. | Simple random. | Paper, entered into database (Excel). | Standalone. | Texas A&M University. | No |
| | Cartagena, Colombia | 2 | University of Miami. | July 2018 | 266 | 69.2 | Dry season. | Spanish. | Simple random. | Paper, entered into database (SPSS). | Standalone. | University of Miami. | No |
| Middle East and North Africa | Beirut, Lebanon | 2 | American University of Beirut. | December 2017–January 2018 | 573 | 63.8 | Rainy season. | Arabic. | Cluster random. | Tablet (ODK). | Standalone. | Northwestern University and American University of Beirut. | Yes |
| | Sistan and Balochistan, Iran | 2 | Shahid Beheshti University of Medical Sciences | January–February 2018 | 306 | 99.0 | Rainy season. | Farsi. | Stratified random. | Paper, entered into database (SPSS). | Standalone; cash transfers and health centre access. | Exempt. | No |
| South Asia | Kathmandu, Nepal | 1 | Arizona State University and Environmental and Public Health Organization. | June 2017 | 263 | 71.5 | Rainy season. | Nepali. | Cluster random. | Paper, entered into database (ODK). | Global Ethnohydrology Study—ASU. | Arizona State University. | Yes |
| | Punjab, Pakistan | 2 | University of Washington. | February–March 2018 | 235 | 57.5 | Dry season. | Seraikee and Urdu. | Cluster random. | Paper, entered into database (Excel). | Standalone; sociocultural feeding practices. | Exempt. | No |
| | Pune, India | 1 | Cornell University, Johns Hopkins University, BJ Government Medical College. | February–present | 180† | 100† | Across multiple seasons. | Marathi and Hindi. | Parallel assignment, non-randomised. | Paper, entered into database (Excel). | NIH R01HD081929: pregnancy associated changes in TB immunology; NIH K23AI129854: effect of pregnancy and HIV on the development of tuberculosis. | Cornell University, Pune IRB, Johns Hopkins University. | No |
| | Rajasthan, India | 2 | Anode Governance Lab. | March 2018 | 248 | 27.0 | Dry season. | Hindi. | Stratified random. | Paper, entered into database (Excel). | Standalone. | Exempt. | No |

*Data were collected for both men and women living in the same household; therefore, for scale analysis, a random individual was selected from each pair (n=564).
†Data collection ongoing, values based on data available as of August 2018.
DRC, Democratic Republic of the Congo; HWISE, Household Water Insecurity Experiences; n.d., no data; ODK, Open Data Kit.

item and survey section (online supplementary material 1). This manual has been translated from English into Arabic and adapted for use in Uganda.

Each site has at least one formally appointed lead investigator responsible for consistent training, sampling, recruitment and data collection. In each site, 5–10 enumerators with survey implementation experience, knowledge of the area and context, and fluency in the local language(s) are recruited. Enumerators at all sites attend a 1–2 day training session. The first portion of the training curriculum is didactic and follows the survey manual. The rest of the training is interactive and tactile, with enumerators piloting the survey with one another and troubleshooting any issues that arise. After the initial training, the site lead and/or study coordinator accompany enumerators during data collection and provide feedback until enumerators are sufficiently comfortable with the survey to administer it with minimal guidance.

### Data collection and management

After consent, enumerators conduct interviews with the person who identifies themselves to the enumerator as being knowledgeable about water acquisition and use in his or her household. In addition to the water insecurity experience items described above (Module Versions 1 or 2), data are collected on sociodemographic characteristics; water acquisition, use and storage; household food insecurity (using the Household Food Insecurity Access Scale[28]); perceived stress (using a modified, four-item perceived stress scale[43]); and data quality (online supplementary material 2). These additional data will be used to validate the scale and explore other water insecurity phenomena in a cross-cultural manner.[44] Each interview lasts approximately 45 minutes, and we expect data collection to last approximately 10–14 days in each standalone survey site (table 3).

Implementation of HWISE data collection began in March 2017 and is expected to end in late 2018. Data collection with Module Version 1 began in March 2017 and is ongoing (table 3, currently n=4817). Data collection using Module Version 2 began in November 2017 and is also ongoing (currently n=3310).

Data are collected using both paper and tablet-based collection platforms, that is, Open Data Kit (ODK), opendatakit.org[45]; CSPro, csprousers.org; and KOBOToolbox (Cambridge, Massachusetts, USA; kobotoolbox.org). To reduce data collection errors, tablet-based platforms are programmed to include permissible ranges of responses, skips for questions that are not applicable and survey items in the language(s) most common to each study site. Most responses from paper surveys are entered by enumerators, study coordinators, data managers and/or site primary investigators into an online data collection platform (Enketo; enketo.org). Microsoft Excel is used when reliable internet access is unavailable.

Data are uploaded to a secure centralised aggregate server (Google App Engine). Stata 14 is used for data cleaning

following a data cleaning protocol agreed on by the HWISE RCN (online supplementary material 3).

### Implementation fidelity

To ensure implementation fidelity, enumerators are debriefed daily following data collection. Both enumerators and site PIs are debriefed postimplementation (online supplementary material 4). Furthermore, each survey contains a module on perceived data quality (eg, explanation of missing data, distractions and issues with recruitment) that is filled in by the enumerator immediately postinterview.

### Analytic strategy

Three software packages will be used for analyses: Stata 14 to run basic descriptive statistics; Mplus version 8 (Muthén & Muthén, Los Angeles, California, USA) and Stata 14 for classical test theory analysis; and WINSTEPS (Winsteps, Beaverton, Oregon, USA) for item response theory (Rasch) analysis.

Scale development (table 1, 2.1–2.4) and evaluation (table 1, 3.1–3.3) will be informed by analyses corresponding to two scaling theories: classical test theory,[46] implemented by factor analysis, and item response theory,[47] using Rasch models.

### Item reduction

First, items with large cumulative missing cases (>30%), that is, 'don't know', 'non-applicable' or true missing responses, will be dropped (table 1, 2.2). This will help to eliminate items that are not understood or are not widely applicable, and therefore do not reflect cross-cultural experiences of water insecurity.

Thereafter, items will be further dropped based on low correlation coefficients. In classical test theory, we will identify items with low (<0.30) interitem and item-total correlation coefficients across the multiple sites in this study.[10 48]

Within the Rasch paradigm, we will identify and remove items that misfit the models by assessing infit and outfit.[49 50] Conditional item independence (ie, items conditional on the scale that are not correlated) will be assessed using residual correlation metrics. Items will be dropped if residual correlation is >0.20.[51]

### Identify factor structure

Factor analysis with data from multiple sites will be used to identify the optimal latent structure (table 1, 2.3). We will examine this structure for each site, comparing factor structures, magnitudes of factor loadings, eigenvalues for sample correlation matrices and global model fitness statistics. Items with low factor loadings (<0.30), split factor loadings and high residual variances (>0.50) will be considered for deletion.[34]

### Assess measurement equivalence

Measurement equivalence concerns the extent to which the psychometric properties of the observed indicators are generalisable across groups or over time.[52–55] It holds 'when a test measures a construct in the same way regardless of group membership and is violated when individuals from

different groups respond to the test in a dissimilar manner'.[39] A violation of equivalence implies our inability to make comparisons about the measurement and meaning of scale values across groups (eg, sites, cultures and languages).[39] To determine measurement equivalence across sites using Module Versions 1 and 2, we will use multigroup confirmatory factor analysis and alignment optimisation.[56–58]

## PHASE 3: SCALE EVALUATION
### Score scale items
Once a water insecurity scale that is equivalent across sites is provisionally identified, we will use scale scores in both weighted forms (factor scores) and unweighted forms (sum scores) to assess the external validity of our scale.

### Reliability
To test for the reliability (internal consistency) of the items, we will estimate Cronbach's alpha for both site-specific and aggregate-level data.[59] The Rasch reliability statistic is analogous to Cronbach's alpha. In our analyses, we will consider reliability to be ideal if it is greater than 0.80.[59]

### Validity
We will examine three types of validity: predictive, convergent and discriminant validity. Predictive validity is 'the extent to which a measure predicts the answers to some other question or a result which it ought to be related with'.[60] Using both linear regression and structural equation models, we will test for predictive validity by regressing HWISE Scale scores on eg, food insecurity, perceived stress and income.

Convergent validity is the 'degree to which scores on a studied instrument are related to measures of other constructs that can be expected on theoretical grounds and accumulated knowledge to be close to the one tapped into by this instrument'.[48] To test for convergent validity, we will assess the relationships between HWISE Scale scores and individual items that have shown to be closely related to the concept of water insecurity. Specifically, we will use linear regression to examine the strength of the relationships between HWISE Scale scores and eg, time to water source, number of trips to water source and amount of money spent purchasing water. Larger correlation and regression coefficients and smaller SD of residuals will be indicative of support for convergent validity.

Discriminant validity is the 'degree to which scores on a studied instrument are differentiated from behavioral manifestations of other constructs'.[48] A test of differentiation between 'known groups' will be conducted using the Student's $t$-test[34 48]; these groups will be based on accumulated knowledge. We will determine the distribution of household water insecurity scores across known groups, eg primary source of drinking water (improved vs unimproved sources), water treatment (treated vs untreated), gender of household head (male vs female) and injuries associated with water acquisition (yes vs no).[4 5 10 24 61] Under the Rasch measurement model, differentiating between known groups will also be conducted using differential item functioning. We will determine whether each scale item performs differently in each of the subgroups. Differential item functioning is attained when the probabilities of an item being endorsed is unequal for the two subgroups.[10 62]

In sum, selection of the set of items to be included in the final scale will be based on several criteria. The criteria for inclusion of an item are: reliable in each site, fits theoretically and empirically with concepts related to water insecurity, has face and content validity in each site, shows equivalent measurement and meaning across sites and contributes to predictive, convergent and discriminant validity in each site.[63] We anticipate that not every item will meet each criterion perfectly, and judgement about tradeoffs of which items to include will be required. These judgements will be made considering the additional criteria of having a diversity of items in the final scale that cover as many facets of water insecurity as reasonably possible. We anticipate that the final scale will have fewer than 20 items, which will reduce the likelihood of participant fatigue and make its widespread application more feasible.

### Midstudy evaluations
In August 2017, 5 months after data collection began in 8 of 16 Module Version 1 sites, HWISE RCN members met at Northwestern University to review and discuss data received to date and thematically sort HWISE items. This led to the reduction and refinement of HWISE for the second wave of survey implementation (Module Version 2), which is being administered across 12 sites (table 2 and 3). In February 2018, HWISE RCN members involved in scale validation met at McGill University to review Module Version 2 responses to date and further refine the survey. Members of the analytic team also hold regular conference calls to review subsequent results and complete the scale validation process.

## ETHICS AND DISSEMINATION
All participants are verbally consented by enumerators in their language of choice using a standardised script (online supplementary material 2). Study activities are reviewed and approved by all appropriate ethical review boards (table 3).

**Author affiliations**
[1]Department of Anthropology, Institute for Policy Research, Northwestern University, Evanston, Illinois, USA
[2]Department of Anthropology, Northwestern University, Evanston, Illinois, USA
[3]Department of Medicine, University of California, San Francisco, California, USA
[4]Center for Research on Population and Health, American University of Beirut, Beirut, Lebanon

[5]School of Human Evolution and Social Change, Arizona State University, Tempe, Arizona, USA
[6]Department of Health Promotion, Education, and Behavior, University of South Carolina, Columbia, South Carolina, USA
[7]Department of Geography, Texas A&M University, College Station, Texas, USA
[8]Institute for Global Food Security, McGill University, Montreal, Québec, USA
[9]Department of Geography, University of Miami, Coral Gables, Florida, USA

**Acknowledgements** We are very grateful to our participants, without whom this scale would not be possible. We would also like to warmly and sincerely thank the field teams for their hard work and dedication to this project: Velly Emina, Victoria Yesufu, Annah Adakhilan, Adekunbi Adejokun, Adebari Adewunmi, Damola Adelakun, Nike Odunaike, Anthony Sekoni, Ramon Babamole and Kayode Badru (Lagos, Nigeria); Prashant Rimal, Sarita Lawaju, Roshna Twanabasu, Renuka Baidhya, Ayaswori Byanju, Menuka Prajapati and Ranju Magar (Kathmandu, Nepal); Andrew Mvula, Wisdom Mwale, Faith Kanyika, Wyson Samata, Fanney Kanyenda and Mcdonad Mpangwe (Lilongwe, Malawi); Maxwell Akosah-Kusi, David Okai Nunoo, Rita Antanah and Michael Nyoagbe (Accra, Ghana); Gulnoza Sharipova, Ganjina Hudoieva, Gulsara Nozirova, Shahobiddin Murodov, Navrasta Shoeva, Nasiba Gadoeva, Markhabo Ibragimova and Vahidova Saodat (Dushanbe, Tajikistan); Milton Marin Morales (San Borja, Bolivia); Wicklife Odhiambo Orero, Judith Atieno Owuour, Philip Otieno Orude, Sylvia Achieng Odhiambo and Kennedy Oduor (Kisumu, Kenya); Daniel Guerrero, Daniela Avila, Kelly Johana Diaz Ceballos, Valentina Giraldo Bohorquez and Pedro Castillo (Honda, Colombia); Moses Mwebaza, Dorren Bamanya, Alines Mpandu, Alex Kazooza, John Ssemwogerer, Olivia Nakamya, Kimbugwe Muhammed, Simon Kyagera, Gerald Ssozi, Solomon Wakida, Matteo Andrea Corsini, Ann Apio, Atim Catu and Nahwera Julie (Kampala, Uganda); Alonzi Francis, Candia Alex, Alesi Christine, Adjonye Doreen, Aputru Florence and Ayakaka Beatrice (Arua, Uganda); Michel Lupamba, Mary Aziza Mulumba, Smith Tshibulenu, Kevis Kamanda and Thérèse Hosa (Kahemba, Democratic Republic of Congo); Eliwaza Mpeko, James Raphael, Emmanuel Katabi, Raziki Amon, Theresia Ononga, Eliofoo Yohana, Neilu Issack, Faudhia Kitiku, Janeth Kacholi, Nyambuli Deus, Oliver Mwanjati, Faith Titus, Fadhali Nyasiro and Mwantum Mkama (Singida, Tanzania); Robinson Bernier, Berlyne Bien-Aimé and Claude Civil (Gressier & Léogâne, Haiti); Luambano Kihoma, Generoza Amos, Patricia Msolla, Peter Amandus, Cyril Lissu, Fredy Bernard, Joylight Mbitta, Raphael Chelele, Alan Kimbita and Elizabeth Msiuike (Morogoro, Tanzania); Daniel Eduardo Lemaitre, Saray Noel Tarra, Luis Murillo Ortega, Natalia Yepes Montes, Jairo Andres Aviles Rojano, Juan Jose de la Espriella Correa, Marcela Florez, Juan Andres Barrios, Stephanie Escobar Diaz, Yuriza Martinez and Carlos Anibal Batista Ruiz (Cartagena, Colombia); Desetaw Asnakew, Roza Abesha, Tigist Abebe and Yeserash Gashu (Bahir Dar, Ethiopia). Furthermore, we are grateful to Northwestern University Information Technology's Research Computing Services team, especially Frank Elavsky for his creation of figure 1.

**Collaborators** HWISE Research Coordination Network: Ellis Adams; Farooq Ahmed; Mallika Alexander; Mobolanle Balogun; Michael Boivin; Genny Carrillo; Kelly Chapman; Stroma Cole; Hassan Eini-Zinab; Jorge Escobar-Vargas; Matthew C. Freeman; Hala Ghattas; Ashley Hagaman; Nicola Hawley; Kenneth Maes; Jyoti Mathad; Patrick Mbullo Owour; Javier Moran; Nasrin Omidvar; Amber Pearson; Asher Rosinger; Luisa Samayoa-Figueroa; Ernesto Sánchez-Rodriguez; Jader Santos; Marianne V. Santoso; Sonali Srivastava; Chad Staddon; Andrea Sullivan; Yihenew Tesfaye; Nathaly Triviño-León; Alex Trowell; Desire Tshala-Katumbay; Raymond Tutu; Felipe Uribe-Salas; Elizabeth Wood; and Cassandra Workman.

**Contributors** SLY conceptualised the study, developed HWISE items, wrote the manuscript, obtained funding and oversaw data collection and analysis. SMC helped develop HWISE items, wrote the manuscript, prepared the field manual and managed data. GOB developed the data analysis and validation plan and helped write the analytic section of the manuscript. TBN assisted with study design and supported scale analysis and validation. ZJ proposed data analysis and helped write the analytic section of the manuscript. JDM developed tools for data collection and managed data. AAB proposed analyses for item development. EAF and HM-Q proposed data analysis. WEJ conceptualized the study and developed HWISE items. RCS supported development of HWISE items and assisted with preparation of the manual. JBS proposed analyses for item development. AW conceptualized the study, developed HWISE items and proposed analyses for item development. HWISE RCN members provided substantial contributions to data acquisition and interpretation. All authors critically reviewed and approved the final draft of the manuscript.

**Funding** We gratefully acknowledge our funders: the Competitive Research Grants to Develop Innovative Methods and Metrics for Agriculture and Nutrition Actions (IMMANA). IMMANA is funded with UK Aid from the UK government. This project was also supported by the Buffett Institute for Global Studies and the Center for Water Research at Northwestern University; Arizona State University's Center for Global Health at the School of Human Evolution and Social Change and Decision Center for a Desert City (National Science Foundation SES-1462086); the Office of the Vice Provost for Research of the University of Miami; the National Institutes of Health grant NIEHS/FIC R01ES019841 for the Kahemba Study, DRC. Texas A&M University-CONACyt Grant supported data collection in Torreon, Mexico. We also acknowledge the National Science Foundation's HWISE Research Coordination Network (BCS-1759972) for support of the collaboration. SLY was supported by the National Institutes of Health (NIMH R21 MH108444; NIMH K01 MH098902). WEJ was supported by the National Science Foundation (BCS-1560962).

**Competing interests** None declared.

**Patient consent** Not required.

**Ethics approval** Northwestern University, African Medical Research Foundation (AMREF), American University at Beirut, Arizona State University, Cornell University, Delaware State University, Florida State University, Georgia State University, Ghana Water Company, International Centre for Diarrhoeal Disease Research, Bangladesh (icddr,b), Johns Hopkins University, College of Medicine at the University of Lagos, University of Miami, McGill University, Michigan State University, Nepal Health Research Council, Oregon Health Sciences University, Oregon State University, Penn State University, Pontificia Universidad Javeriana, Sokoine University of Agriculture, Texas A&M University, T-Group Kampala and Yale University.

**Provenance and peer review** Not commissioned; externally peer reviewed.

**Data sharing statement** Data are currently being collected and are not yet available for access.

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
