## [Reviewer comments · BMJ Open]

ARTICLE DETAILS

TITLE (PROVISIONAL)	Development and validation protocol for an instrument to measure household water insecurity across cultures and ecologies: the Household Water InSecurity Experiences (HWISE) Scale
AUTHORS	Young, Sera; Collins, Shalean; Boateng, Godfred; Neilands, Torsten; Jamaluddine, Zeina; Miller, Joshua; Brewis, Alex; Frongillo, EA; Jepson, Wendy; Melgar-Quiñonez, Hugo; Schuster, Roseanne; Stoler, JB; Wutich, Amber

VERSION 1 – REVIEW

REVIEWER	Edward G. J. Stevenson Durham University, United Kingdom
REVIEW RETURNED	13-Jul-2018

GENERAL COMMENTS	This document lays out the protocol for the largest international effort so far to develop metrics of household water insecurity. Efforts to systematically compare HH WI across settings have not previously been carried out, and this protocol will therefore be of interest to many scholars and practitioners. My comments focus on issues that are important for replicability: regarding issues of site selection, the rationale for the use of particular variables for tests of validity, and the comparison groups used in tests of differentiation. First I point out some ways in which explanation of the protocol as a whole could be made clearer. To ensure the protocol is replicable, the study design and the various stages of field research and analytical work involved need to be explained more clearly. Under 'item generation' (p. 7) 3 successive rounds of data collection are described, but it's unclear why certain sites were focused on in round 1, vs. 2 or 3. The term 'rounds' suggests repeat surveys, but elsewhere it seems the design is cross-sectional. Readers may also find it confusing that, close together in the text, there are references to both the three "rounds" of data collection (or versions of the survey?), and also three "phases" (domain identification, scale dev't, and scale evaluation). The presentation of material needs to be revised here — particularly since "Phase 1" is allocated more than 10 pages of text with many intermediary headings, while Phases 2 and 3 are dispensed with in a few pages towards the end. (For information on the "rounds of data collection" readers are referred to SI describing the items used in each round; an account in the main text of the differences between them would also be useful.)
--

	The authors should clarify the way study locations (countries, and locales within them) were selected. For example, why did they choose the first 6 sites? And what guided them in the choice of sites included later on? While there is appropriate attention to sampling *within* study sites, the countries and locations included as sites in the first place seems to approximate a convenience sample — places where research opportunities presented themselves. Was any effort made to sample for representativeness across world regions, for example, or across areas with different types of water infrastructures? On p. 17 it emerges that the lead authors' earlier work in Kenya was important to the approach taken to ascertaining content validity. It might be appropriate to flag earlier, in the Introduction, the progression from work in Kenya to the international program. More information on the Delphi method and how it was adapted for the international project would also be useful. Some clarifications would be helpful in the section on Scale Evaluation: Under 'Validity' (p. 20) are the "individual items that have shown to be closely related to the concept of water insecurity" determined on the basis of the HWISE study data, or on the basis of the WI literature more generally? And why is "total amount of water stored" chosen as the variable representing a construct other than water insecurity? What is the construct that this variable is taken to stand for? Regarding the tests of differentiation (p. 20-21), it's unclear which groups are referred to by the phrase "known groups". This may be inevitable if the groups are yet to be identified, but (for the sake of replicability) readers need to be told *how* they will be identified. The authors say they will determine these groups based on "theoretical evidence and accumulated knowledge" — which particular bodies of theory and knowledge will they draw on? In the Abstract the claim is made that "measures to quantify household food security have been transformative for policy, research and humanitarian aid efforts globally." The promise of the HWISE project rests in large part on this precedent. This claim about FI measures is repeated twice in the body of the ms. (in the Intro and Conclusion), but without citation. Given the importance of the statement for the HWISE project, it would be good to see references added on this. Finally, the section on Ethics repeats material (the location of IRBs) that is already given in Table 3. This seems unnecessary. This document lays out the protocol for the largest international effort so far to develop metrics of household water insecurity. Efforts to systematically compare HH WI across settings have not previously been carried out, and this protocol will therefore be of interest to many scholars and practitioners. My comments focus on issues that are important for replicability: regarding issues of site selection, the rationale for the use of
--	---

	particular variables for tests of validity, and the comparison groups used in tests of differentiation. First I point out some ways in which explanation of the protocol as a whole could be made clearer. To ensure the protocol is replicable, the study design and the various stages of field research and analytical work involved need to be explained more clearly. Under 'item generation' (p. 7) 3 successive rounds of data collection are described, but it's unclear why certain sites were focused on in round 1, vs. 2 or 3. The term 'rounds' suggests repeat surveys, but elsewhere it seems the design is cross-sectional. Readers may also find it confusing that, close together in the text, there are references to both the three "rounds" of data collection (or versions of the survey?), and also three "phases" (domain identification, scale dev't, and scale evaluation). The presentation of material needs to be revised here — particularly since "Phase 1" is allocated more than 10 pages of text with many intermediary headings, while Phases 2 and 3 are dispensed with in a few pages towards the end. (For information on the "rounds of data collection" readers are referred to SI describing the items used in each round; an account in the main text of the differences between them would also be useful.) The authors should clarify the way study locations (countries, and locales within them) were selected. For example, why did they choose the first 6 sites? And what guided them in the choice of sites included later on? While there is appropriate attention to sampling *within* study sites, the countries and locations included as sites in the first place seems to approximate a convenience sample — places where research opportunities presented themselves. Was any effort made to sample for representativeness across world regions, for example, or across areas with different types of water infrastructures? On p. 17 it emerges that the lead authors' earlier work in Kenya was important to the approach taken to ascertaining content validity. It might be appropriate to flag earlier, in the Introduction, the progression from work in Kenya to the international program. More information on the Delphi method and how it was adapted for the international project would also be useful. Some clarifications would be helpful in the section on Scale Evaluation: Under 'Validity' (p. 20) are the "individual items that have shown to be closely related to the concept of water insecurity" determined on the basis of the HWISE study data, or on the basis of the WI literature more generally? And why is "total amount of water stored" chosen as the variable representing a construct other than water insecurity? What is the construct that this variable is taken to stand for? Regarding the tests of differentiation (p. 20-21), it's unclear which groups are referred to by the phrase "known groups". This may be inevitable if the groups are yet to be identified, but (for the sake of replicability) readers need to be told *how* they will be identified. The authors say they will determine these groups based on "theoretical evidence and accumulated knowledge" — which particular bodies of theory and knowledge will they draw on?
--	---

	In the Abstract the claim is made that “measures to quantify household food security have been transformative for policy, research and humanitarian aid efforts globally.” The promise of the HWISE project rests in large part on this precedent. This claim about FI measures is repeated twice in the body of the ms. (in the Intro and Conclusion), but without citation. Given the importance of the statement for the HWISE project, it would be good to see references added on this. Finally, the section on Ethics repeats material (the location of IRBs) that is already given in Table 3. This seems unnecessary.
--	---

REVIEWER	Christine Stauber Georgia State University, United States
REVIEW RETURNED	06-Aug-2018

GENERAL COMMENTS	I think that this paper describes an important activity for reviewing, defining and implementing a measure of household water insecurity across varied settings. I think the protocol approach merits publication and I have some general and then some specific comments for the authors to address. 1) The authors state that the development of food insecurity measures have been transformed that field. Please provide examples. In addition, it would be important to describe in more detail who might the target audience or audiences both for the implementation of this HWISE scale and for the results from it. 2) Along the lines of the comment above, it seems that the authors missed an opportunity to consider how this scale fits in with the indicators and measures related to the Sustainable Development Goals for water and sanitation. https://sustainabledevelopment.un.org/sdg6 The following comments are more specific: (page number refers to pdf page and not manuscript page). In the methods section, on page 8, line 34- can you clarify how expert input was received and used to modify this information? How were experts selected? Was there a systematic process for this? Who were the experts, etc. On page 9 (line 3-10), you describe the frequency experienced as rarely, sometimes, etc? What is the basis for determination of these cut-points? For example, how much different is rarely than sometimes in terms of the impact on the household or individual. What is the basis for these cut-points? More details (even if provided in supplemental materials) should be included here. As a follow up, you mention a range of participants in the consortium (page 6, line 50) and their expertise but is there a statistician engaged with the group? On page 9, line 14, please clarify how the first six sites were selected and why? On page 17, line 9 you discuss training and implementation. Beyond data cleaning and analysis, Is there an assessment of implementation fidelity?
---

	Lastly, after reviewing the survey presented in the supplemental materials, is there any discussion of modifying the length of the survey?
--	--

VERSION 1 – AUTHOR RESPONSE

Reviewers' Comments to Author:

Reviewer: 1

Reviewer Name: Edward G. J. Stevenson

Institution and Country: Durham University, United Kingdom

Competing Interests: None declared

This document lays out the protocol for the largest international effort so far to develop metrics of household water insecurity. Efforts to systematically compare HH WI across settings have not previously been carried out, and this protocol will therefore be of interest to many scholars and practitioners.

* Thank you.

My comments focus on issues that are important for replicability: regarding issues of site selection, the rationale for the use of particular variables for tests of validity, and the comparison groups used in tests of differentiation. First I point out some ways in which explanation of the protocol as a whole could be made clearer.

* These are all appreciated.

To ensure the protocol is replicable, the study design and the various stages of field research and analytical work involved need to be explained more clearly. Under 'item generation' (p. 7) 3 successive rounds of data collection are described, but it's unclear why certain sites were focused on in round 1, vs. 2 or 3.

* Because there is only 1 item difference between version 2.0 and 3.0 of the survey, we have collapsed 2.0 and 3.0 for simplicity. The evolution of 1.0 and 2.0 versions of the scale are now explained in the text at lines 156-163 and illustrated in Supplemental Material 1.

The term 'rounds' suggests repeat surveys, but elsewhere it seems the design is cross-sectional. Readers may also find it confusing that, close together in the text, there are references to both the three "rounds" of data collection (or versions of the survey?), and also three "phases" (domain identification, scale dev't, and scale evaluation). The presentation of material needs to be revised here — particularly since "Phase 1" is allocated more than 10 pages of text with many intermediary headings, while Phases 2 and 3 are dispensed with in a few pages towards the end. (For information on the "rounds of data collection" readers are referred to SI describing the items used in each round; an account in the main text of the differences between them would also be useful.)

*Your suggestion to distinguish rounds and phases is well-taken. We have streamlined to use the language "HWISE 1.0, 2.0" to refer to which version of scale items we were asking about. We have dropped the use of 'rounds' completely. "Phases" is used to refer to phases in scale development, as laid out in Table 1, and consistently referred to throughout the text. We have now described changes between HWISE 1.0 and 2.0 at lines 156-163, as well as in Supplemental Material 1.

To your second point-- it is true that the first phase of scale development receives much more attention and space here than phases 2 and 3. Those are more analytic in nature, i.e. rely on data and will be published in due course in the final scale paper. Because phases 2 and 3 of scale development require the completed dataset, and because study activities are ongoing, we anticipate presenting these results once data collection and analyses are complete.

The authors should clarify the way study locations (countries, and locales within them) were selected. For example, why did they choose the first 6 sites? And what guided them in the choice of sites included later on? While there is appropriate attention to sampling *within* study sites, the countries and locations included as sites in the first place seems to approximate a convenience sample — places where research opportunities presented themselves. Was any effort made to sample for representativeness across world regions, for example, or across areas with different types of water infrastructures?

* The 28 sites were indeed selected by convenience. The initial 6 were selected because the investigators had active research there. However, we suspected that the range of water access experiences were not captured in these sites. Thus, we then turned to our networks to identify other sites that were amenable to the HWISE survey being administered. We wanted to be sure that there were sites in each of the LMIC WB regions, and to include sites in areas with varied typologies of water insecurity such as flooding, chronic insufficiency, drought, and intermittent supplies, among others. We also sampled to capture rural and urban areas and differences in infrastructure (e.g. open sources, communal taps, water piped into homes, etc.). This is now explained in the text at lines 191-201.

On p. 17 it emerges that the lead authors' earlier work in Kenya was important to the approach taken to ascertaining content validity. It might be appropriate to flag earlier, in the Introduction, the progression from work in Kenya to the international program.

* We mention earlier, site specific scales created by co-authors and others at lines 97 and 140-143.

More information on the Delphi method and how it was adapted for the international project would also be useful.

* The Delphi Method was described by Boateng et al., PLoS One 2018. A novel household water insecurity scale: Procedures and psychometric analysis among postpartum women in western Kenya. This is referenced at line 149 and 155.

Further, Supplemental Table 1 of the Plos One article includes all questions used in our implementation of this methodology:

“C. Delphi method. The Delphi method is a technique ^afor structuring a group communication process so that the process is effective in allowing a group of individuals, as a whole, to deal with a complex problem^o [50]. Here, it was used to obtain feedback from international experts including those with expertise in hydrology and geographic research, WASH and water related programs, policy implementation, food insecurity and scale development, over the course of three rounds of surveys (S1 Fig). Each round was interspersed with FGDs in which questionnaires progressively became more closed ended. Questions included the definition of water insecurity, household water-related activities, barriers to water acquisition, consequences of water insecurity, and possible survey items that could constitute a HHWI scale (S1 Table).

D. Focus group discussions. FGDs were conducted iteratively with the Delphi process (S1 Fig). To participate in FGDs, nurses and healthcare professionals purposively recruited postpartum women who were available and were either pregnant or had children less than 2 years of age in 4 study areas. After Delphi round 1, FGD participants (Kisumu; n = 8 and Rongo; n = 7) were asked to provide feedback on topics discussed in the online survey to build consensus around the definition of and questions related to HHWI. Another group of FGD participants (Migori; n = 5 and Macalder; n = 7) also provided information with which to revise questions for the survey.”

Some clarifications would be helpful in the section on Scale Evaluation:

Under 'Validity' (p. 20) are the “individual items that have shown to be closely related to the concept of water insecurity” determined on the basis of the HWISE study data, or on the basis of the WI literature more generally? And why is “total amount of water stored” chosen as the variable representing a construct other than water insecurity? What is the construct that this variable is taken to stand for?

* The items closely related to the concept of water insecurity are determined based on water insecurity literature more generally. In the absence of constructs, items that have shown to be related to water insecurity will be used.

Regarding the tests of differentiation (p. 20-21), it's unclear which groups are referred to by the phrase "known groups". This may be inevitable if the groups are yet to be identified, but (for the sake of replicability) readers need to be told *how* they will be identified. The authors say they will determine these groups based on "theoretical evidence and accumulated knowledge" — which particular bodies of theory and knowledge will they draw on?

* We have more details about added known groups at lines 365-368:

"We will determine the distribution of household water insecurity scores across known groups, including primary source of drinking water (improved vs. unimproved sources), water treatment (treated vs. untreated), gender of household head (male vs. female), and injuries associated with water acquisition (yes vs. no)."

These groups were constructed using accumulated knowledge, and we have removed "theoretical evidence." We have also included references that were informative in selecting these groups, particularly:

Boateng et al. (2018). A Novel Household Water Insecurity Scale: Procedures and Psychometric Analysis Among Postpartum Women in Western Kenya.

Wutich & Ragsdale. (2008). Water insecurity and emotional distress: coping with supply, access, and seasonal variability of water in a Bolivian squatter settlement.

Tsai et al. (2015). Population-based study of intra-household gender differences in water insecurity: reliability and validity of a survey instrument for use in rural Uganda.

Stevenson et al. (2012). Water insecurity in 3 dimensions: an anthropological perspective on water and women's psychosocial distress in Ethiopia.

Hadley & Freeman (2016). Assessing reliability, change after intervention, and performance of a water insecurity scale in rural Ethiopia.

In the Abstract the claim is made that "measures to quantify household food security have been transformative for policy, research and humanitarian aid efforts globally." The promise of the HWISE project rests in large part on this precedent. This claim about FI measures is repeated twice in the body of the ms. (in the Intro and Conclusion), but without citation. Given the importance of the statement for the HWISE project, it would be good to see references added on this.

* We have removed this from the Introduction to avoid repetition and have included citations that support these claims at line 110.

Finally, the section on Ethics repeats material (the location of IRBs) that is already given in Table 3. This seems unnecessary.

* We have removed the text and retained the information in Table 3.

Reviewer: 2

Reviewer Name: Christine Stauber

Institution and Country: Georgia State University, United States

Competing Interests: None declared

I think that this paper describes an important activity for reviewing, defining and implementing a measure of household water insecurity across varied settings. I think the protocol approach merits publication and I have some general and then some specific comments for the authors to address.

* Thank you!

The authors state that the development of food insecurity measures have been transformed that field.

Please provide examples.

* We have removed the text from the Introduction to avoid repetition, and have included citations that support these claims at line 110.

In addition, it would be important to describe in more detail who might the target audience or audiences both for the implementation of this HWISE scale and for the results from it.

* We have revised at lines 100-105 to read:

“As such, a comprehensive, validated scale to measure the experiences of household or individual water insecurity would enable researchers, practitioners, and policymakers to: improve estimates of water insecurity prevalence, identify factors that shape this phenomenon, recognize direct consequences of water insecurity, understand how to more effectively distribute resources, evaluate the impacts and cost-effectiveness of interventions, and monitor progress toward the Sustainable Development Goals.”

2) Along the lines of the comment above, it seems that the authors missed an opportunity to consider how this scale fits in with the indicators and measures related to the Sustainable Development Goals for water and sanitation. https://urldefense.proofpoint.com/v2/url?u=https-3A__sustainabledevelopment.un.org_sdg6&d=DwIFaQ&c=yHIS04HhBraes5BQ9ueu5zKhE7rNXt_d012z2PA6ws&r=s7dUTPTapOslnCH82qIWHJUUXSa0J-VaUSWUYBz4Jnc&m=4Omh9kLwTNh5hIKn0vQNN_kHxaAauxxUTbYyxTs_OQ4&s=BMjjiPtrB8tcD83E1ufZomE6OuXuyphWVRqHYeUsk8A&e=

* This is a terrific idea. We now comment on this, and also link the potential value of this scale to the call for data made by the UN's High Level Panel on Water at lines 105-107.

The following comments are more specific:

(page number refers to pdf page and not manuscript page). In the methods section, on page 8, line 34- can you clarify how expert input was received and used to modify this information? How were experts selected? Was there a systematic process for this? Who were the experts, etc.

* We have revised this at lines 156-163:

“The initial set of 32 items is referred to as “HWISE 1.0”. This set of items was modified slightly in August 2017 (cf. “Mid-study Evaluations”) based on feedback received from consortium members, survey implementers, and other water security experts during a three-day conference at Northwestern University. Modifications included slight rephrasing of 18 items to improve comprehension by participants and elicit experiences related to water overabundance, two questions were added in an effort to capture cultural components of water, and six items were eliminated for being too rare or idiosyncratic. The resultant set of 28 items is referred to as “HWISE 2.0” (Supplemental Material 1: HWISE 1.0 and 2.0 items).”

On page 9 (line 3-10), you describe the frequency experienced as rarely, sometimes, etc? What is the basis for determination of these cut-points? For example, how much different is rarely than sometimes in terms of the impact on the household or individual. What is the basis for these cut-points? More details (even if provided in supplemental materials) should be included here.

* We have revised this at lines 148-155:

“This recall period was systematically determined using the Delphi method of consensus building with international and local experts in water insecurity, food insecurity, and scale development, and by comparing the responses in this recall period to a prospective daily recall of water insecurity experiences. Items are ordered by what we expect to be increasing severity of water insecurity across access, reliability, adequacy, and safety. Response options are “never” (0 times), “rarely” (1-2 times), “sometimes” (3-10 times), “often” (11-20 times), “always” (more than 20 times), “not applicable”, “don't know”, or refused. Response intervals were also determined using the Delphi method.”

As a follow up, you mention a range of participants in the consortium (page 6, line 50) and their expertise but is there a statistician engaged with the group?

* Yes, several. This has been added, and the statisticians are also highlighted on the website (<http://sites.northwestern.edu/hwise/collaborators/>).

On page 9, line 14, please clarify how the first six sites were selected and why?

* The 28 sites were indeed selected by convenience. The initial 6 were selected because the investigators had active research there. However, we suspected that the range of water access experiences were not captured in these sites. Thus, we then turned to our networks to identify other sites that were amenable to the HWISE survey being administered. We wanted to be sure that there were sites in each of the LMIC WB regions, and to include sites in areas with varied typologies of water insecurity such as flooding, chronic insufficiency, drought, and intermittent supplies, among others. We also sampled to capture rural and urban areas and differences in infrastructure (e.g. open sources, communal taps, water piped into homes, etc.). This is now explained in the text at lines 191-201.

On page 17, line 9 you discuss training and implementation. Beyond data cleaning and analysis, is there an assessment of implementation fidelity?

* We have revised this at lines 292-296:

“To ensure implementation fidelity, enumerators are debriefed daily following data collection. Both enumerators and site PIs are debriefed post-implementation. Further, each survey contains a module on perceived data quality (e.g. explanation of missing data, distractions, and issues with recruitment) that is filled in by the enumerator immediately post-interview.”

Lastly, after reviewing the survey presented in the supplemental materials, is there any discussion of modifying the length of the survey?

* We hope to reduce the number of items in the HWISE module well below 20. This is stated at lines 380-382.

Additionally, we have made several other edits:

1. We added sites that have collected data since the review of this manuscript, including Torreón, Mexico; Sistan & Balochistan, Iran; Cartagena, Colombia. We have also removed Kisoro, Uganda and Bangladesh due to time constraints that prevented data collection in these sites.
2. We have made minor updates to the tables based on feedback received from collaborators and updated sample sizes.
3. We have added the end date of data collection activities in each of the sites where data collection has ended, and have presented the proportion of female respondents for each site.
4. We have added consortium authorship. This change is appropriate because we have added gender disaggregated responses by site, i.e. data that was collected by many who had previously been named in acknowledgements, but not named as authors. Further, many of those same individuals made substantive comments resulting in edits to the water insecurity experiences questions. I have ensured that all consortium authors meet the ICJME guidelines for authorship, and have their signatures.

VERSION 2 – REVIEW

REVIEWER	Edward G. J. Stevenson Durham University, United Kingdom
REVIEW RETURNED	12-Oct-2018

GENERAL COMMENTS	My comments are few and should be easy to rectify. In terms of style, there is inconsistency between use of present, past, and future tense to describe different parts of the study. It's unclear whether these relate consistently to parts that have already been done, are underway, or are planned. "There are no plans to disseminate the final scale to study participants, as identifiable data were not collected in most sites." (p. 16). Good practice of sharing data with study participants does not, in my view, rest on whether or not they are identifiable in study dataset / outputs. I didn't catch this first time around, but it seems strange that there is no data on climate type for Samoa, in Table 2
--

VERSION 2 – AUTHOR RESPONSE

In terms of style, there is inconsistency between use of present, past, and future tense to describe different parts of the study. It's unclear whether these relate consistently to parts that have already been done, are underway, or are planned.

** The verb tense in the protocol is indeed tricky, because some steps have been done, some are ongoing, and some still to occur. In short, item development and construct validity have been completed ("Phase 1," 1.1-1.3 on Table 1), face validity and data collection are ongoing (1.4 and 2.1 on Table 1), and scale evaluation is upcoming (2.2 onwards on Table 1). Accordingly, throughout the manuscript, we have revised verb tenses to reflect past, present, and future study activities. If your copy editor suggests any revisions to this, we are quite open to further modification.

"There are no plans to disseminate the final scale to study participants, as identifiable data were not collected in most sites." (p. 16). Good practice of sharing data with study participants does not, in my view, rest on whether or not they are identifiable in study dataset / outputs.

** We have clarified that our plan is to summarize our findings in site-specific reports that investigators will disseminate to community leaders (lines 282-284). However, many sites did not collect identifiable information, so we are unable to directly share findings with most study participants. We can, however, share findings with the communities themselves.

I didn't catch this first time around, but it seems strange that there is no data on climate type for Samoa, in Table 2.

** Unfortunately, Köppen climate classification data do not exist for Samoa (<https://bit.ly/2Eu78C9>).